# *n*-6 Linoleic Acid Induces Epigenetics Alterations Associated with Colonic Inflammation and Cancer

**DOI:** 10.3390/nu11010171

**Published:** 2019-01-15

**Authors:** Donato F. Romagnolo, Micah G. Donovan, Tom C. Doetschman, Ornella I. Selmin

**Affiliations:** 1The University of Arizona Cancer Center, Tucson, AZ 85724, USA; selmin@email.arizona.edu; 2Department of Nutritional Sciences, University of Arizona, Tucson, AZ 85721, USA; 3Interdisciplinary Cancer Biology Graduate Program, University of Arizona, Tucson, AZ 85724, USA; mdono123@email.arizona.edu; 4Department of Cellular & Molecular Medicine, University of Arizona, Tucson, AZ 85724, USA; tdoetsch@email.arizona.edu

**Keywords:** *n*-6 linoleic acid, high-fat diet (HFD), bile acids, epigenetics, farnesoid-X-receptor (FXR), cyclooxygenase-2 (COX-2), *Apc*, colon cancer

## Abstract

The farnesoid-X-receptor (FXR) protects against inflammation and cancer of the colon through maintenance of intestinal bile acid (BA) homeostasis. Conversely, higher levels of BA and cyclooxygenase-2 (COX-2) are risk factors for inflammation and cancer of the colon. In the United States, *n*-6 linoleic acid (LA) is the most commonly used dietary vegetable fat. Metabolism of *n*-6 fatty acids has been linked to a higher risk of intestinal cancer. The objectives of this study were to investigate in colonic mucosa the effects of a high-fat diet rich in LA (*n*-6HFD) on CpG methylation of *Fxr* and prostaglandin-endoperoxide synthase-2 (*Ptsg-2*) genes, and the impact on the expression of tumor suppressor adenomatous polyposis Coli (*Apc*) and proliferative cyclin D1 (*Ccnd1*) genes. Weaned C57BL/6J male mice were fed for 6 weeks either an *n*-6HFD containing 44% energy (44%E) from 22% safflower oil (SO, 76% LA by weight) or a 13% energy (13%E) control diet (Control) from SO (5% by weight). Mice fed the *n*-6HFD had reduced (60%) *Fxr* promoter CpG methylation and increased (~50%) *Fxr* mRNA. The expression of FXR-target ileal bile acid-binding protein (*Ibabp*), small heterodimer protein (*Shp*), and anti-inflammatory peroxisome proliferator-activated-γ1 genes was increased. The *n*-6HFD reduced *Ptgs-2* CpG methylation, increased the expression of *Cox-2*, and increased *Apc* CpG methylation in colonic mucosa. Accordingly, reduced expression of *Apc* was coupled to accumulation of c-JUN and *Ccnd1,* respectively cofactor and gene targets for the β-catenin/Wnt signaling pathway. Finally, the *n*-6HFD reduced the expression of histone deacetylase-1 while favoring the accumulation of acetylated histone 3. We conclude that an *n*-6HFD epigenetically modifies *Fxr*, leading to the activation of downstream factors that participate in BA homeostasis. However, epigenetic activation of *Ptsg-2* coupled with silencing of *Apc* and accumulation of C-JUN and *Ccnd1* may increase the risk of inflammation and cancer of the colon.

## 1. Introduction

A diet high in *n*-6 fatty acids (*n*-6HFD) is a recognized risk factor for inflammatory bowel diseases (IBDs) [1,2]. Individuals with diets high in linoleic acid (LA), an *n*-6 fatty acid found predominantly in plant oils (e.g., soybean, corn, safflower), are at increased risk for one common form of IBD, ulcerative colitis [3,4]. In animal studies, the feeding of a vegetable oil (corn oil) rich in LA increases cell proliferation in the colon [5,6]. An *n*-6HFD induces the biliary excretion of bile acids (BA), increases fecal levels of the secondary BA deoxycholic acid (DCA), and chemically induced colon cancer incidence in rodent models [7]. A causative relationship exists between colon cancer development and a high concentration of rectal BA in preclinical models [8] and human patients [9,10,11,12]. Because colon rectal cancer (CRC) is a leading cause of death and its incidence in early onset patients (<55 years of age) is on the rise [13], there is great interest in clarifying whether or not intake of an *n*-6HFD increases the risk of CRC through epigenetic mechanisms [14,15,16].

The farnesoid-X-receptor (FXR) coordinates the expression of genes encoding for enzymes involved in BA homeostasis through the enterohepatic circulation. These include ileal bile acid-binding protein (IBABP) and small heterodimer protein (SHP). In the intestine, SHP lowers the expression of the sodium-dependent bile acid transporter, whereas IBABP coordinates the transcellular movement of BA to the basolateral membrane [17]. In humans, the loss of FXR expression is noted during the transition from late stage colon adenoma to carcinoma [18], and correlates with a higher tumor grade and a poor clinical pathological response [19]. In rodent models, overexpression of *Fxr* ameliorates tumor growth [20], whereas the genetic deletion of *Fxr* augments chemically induced colon tumorigenesis [21]. On the other hand, cyclooxygenase-2 (COX-2) overexpression is linked to increased production from arachidonic acid (AA) of prostaglandin E2 (PGE2), which induces cell proliferation [22] and development of CRC [23]. The increased expression of COX-2 and nuclear translocation of β-catenin are related to a loss of expression of the adenomatous polyposis Coli (*Apc*) gene and intestinal tumorigenesis [24]. Therefore, dietary factors that alter the relative balance between tumor protective (FXR, *Apc*) and promoting (COX-2) factors may disrupt homeostasis and contribute to intestinal inflammation and cancer. In this study, we investigated in a mouse model the effects of an *n*-6HFD rich in LA on CpG methylation of *Fxr*, prostaglandin-endoperoxide synthase-2 (*Ptsg-2*), and *Apc* genes, and the expression of factors involved in BA homeostasis (*Ibabp* and *Shp*), proliferation (*Ccnd1*), and oncogenic transformation (c-JUN). Our findings provide new mechanistic insights into the role of an *n*-6HFD as a risk factor for CRC through epigenetic activation of *Ptsg-2* and silencing of *Apc*.

## 2. Materials and Methods

### 2.1. Animal Models

Weaned C57/BL6 male mice were purchased from Jackson Laboratories (Bar Harbor, ME, USA) and assigned to a control diet containing 13% energy (13%E, 5% by weight safflower oil (SO), 76% LA) or an *n*-6HFD containing 44% energy (44%E, 22% by weight SO) (Harlan Laboratories, Madison, WI, USA) for 6 weeks (Table 1). Animals were allowed chow and water ad libitum, and their weight was recorded twice a week. At the end of the experimental periods, colonic tissue was collected as described previously [25]. Briefly, the large intestine was cut open longitudinally and rinsed with ice-cold phosphate-buffered saline (PBS). The proximal colonic mucosa was scraped and cells separated by centrifugation. All animal procedures were approved by the Institutional Animal Care and Use Committee (IACUC) program of The University of Arizona.

### 2.2. Cell Lines and Reagents

Human colonic fetal human cells (FHC) were obtained from the American Type Culture Collection (Manassas, VA, USA) and maintained in Dulbecco’s Modified Eagle’s Medium (DMEM) from Sigma-Aldrich (St. Louis, MO, USA) supplemented with 10% fetal calf serum (FCS) (Hyclone Laboratories, Logan, UT, USA) as described previously [26]. Linoleic acid and DCA were purchased from Sigma-Aldrich. At the end of the treatment period, cells were rinsed with PBS, precipitated by centrifugation, and stored at −80 °C until further analysis. Western blotting was performed using antibodies for COX-2 (Cayman Chemical, Ann Arbor, MI, USA); c-JUN (Cell Signaling Technology, Danvers, MA, USA); histone deacetylase-1 (HDAC-1) and acetylated histone-3 (ACH3) (MilliporeSigma, Burlington, MA, USA); and FXR and β-ACTIN (Santa Cruz Biotechnology, Dallas, TX, USA).

### 2.3. Real-Time PCR

Colonic mucosa were scraped from the proximal colon as described previously [27], and total RNA was prepared using the Quick-RNA Miniprep Kit (Zymo Research, Irvine, CA, USA). The RNA concentration was assessed with the Nanodrop1000 Spectrophotomoter (Thermo Scientific, Waltham, MA, USA). Total RNA (500 ng) was used to prepare cDNA using qScript cDNA SuperMix (Quanta Biosciences, Gaithersburg, MD, USA). For PCR amplification of cDNA, we used the PerfeCTa SYBR Green Fast Mix, Rox (Quanta Biosciences). PCR reactions were performed at a final volume of 10 μL comprising 5 μL of SYBR Green Mix, forward and reverse primers (1 μL each, 10 nM), 2 μL nuclease-free water, and 1 μL of cDNA. Glyceraldehyde 3-phosphate dehydrogenase (*Gapdh*) amplification was used for normalization of mRNA expression. The mouse primers (Sigma-Aldrich) used for RT-PCR are shown in Table 2.

### 2.4. Genomic DNA Methylation

The procedure for measurement of promoter methylation was described previously [26]. In short, genomic DNA was prepared from 10–15 mg of proximal colon mucosa using the DNeasy Blood and Tissue Kit (Qiagen, Valencia, CA, USA). The DNA (1 μg) was modified via bisulfite treatment using the EpiTect Bisulfite Modification Kit (Qiagen) followed by PCR amplification using 1 cycle at 94 °C (1 min); 33–35 cycles at 94 °C (30 s), 59 °C (30 s), and 72 °C (1 min); and 1 cycle at 72 °C (5 min). Amplification was performed at a volume of 25 μL comprising bisulfite modified genomic DNA (50 ng), 0.4 μL JumpStart Taq DNA polymerase (Sigma-Aldrich), 2.5 μL 10X PCR buffer, 3.5 μL 25 mM MgCl_2_ (final concentration 3.5 mM), 0.5 μL 10 mM dNTP mix (final concentration 200 μM), forward and reverse primers (1 μL each), and water to a final volume of 25 μL. The PCR products were analyzed on 2% agarose gels and examined by ethidium bromide staining. The size and authenticity of the PCR products were confirmed by a molecular weight analysis and DNA sequencing. The primers (Sigma-Aldrich) used for DNA methylation studies are shown in Table 2.

### 2.5. Statistical Analysis

Densitometry analyses of PCR products were carried out using the Kodak ID Image Analysis Software (Eastman Kodak Company, Rochester, NY, USA). Expression and promoter methylation data were analyzed by ANOVA. Post-hoc multiple comparisons among all means were performed using Tukey’s test after the main effects and interactions were confirmed to be significant at *p* ≤ 0.05. Data are presented as means ± standard error of the mean (SEM) and statistical differences highlighted with asterisks or different letters for multiple comparisons.

## 3. Results

### 3.1. n-6HFD Induces Fxr Gene Promoter Hypomethylation and Expression in Proximal Colonic Mucosa

Compared to animals assigned to the control diet (5%E), mice fed the *n*-6HFD (44%E) had higher weight gain (~10%) starting at week 4 through week 6 of the experiment (Figure 1A).

We examined the effects of the *n*-6HFD on *Fxr* mRNA expression in proximal colonic mucosa and found that *Fxr* transcripts were increased by ~40% compared to animals fed the control diet (Figure 1B). As a control for changes in regulation of lipid metabolism, we monitored the expression of *Pparγ1,* which was increased by ~80%. The upregulation of *Fxr* and *Pparγ1* were consistent with findings of previous reports documenting higher weight gain and increased levels of FXR in the distal small intestine in response to a high-fat diet (HFD) [28], and FXR-mediated transcriptional activation of *Pparγ* [29]. We then turned our analysis to the intestinal gene targets for FXR, *Ibabp*, and *Shp*, whose expression was increased respectively 1.5- and 3.2-fold compared to the control (Figure 1C,D). Based on these results, we examined the impact of the *n*-6HFD on the CpG methylation status of *Fxr*. We focused on a 470 bp promoter region (−54/+416) comprising 13 CpG sites and flanking the transcription start site (+1) of exon-3. This promoter generates FXRα3/4 transcripts (Figure 2A), which are expressed at higher levels than the FXRα1/2 mRNA variants in the intestine [20].

Amplification of CpG-methylated *Fxr* promoter amplicons from bisulfonated genomic colonic mucosal DNA was conducted in the linear range as reported previously [26]. The *Fxr* gene CpG methylation was reduced by ~60% (Figure 2B,C) in mice fed the *n*-6HFD, in agreement with a previous experimental report documenting increased intestinal FXR expression and BA levels as a result of feeding an HFD [30].

### 3.2. n-6HFD Induces Ptsg-2 Gene Promoter Hypomethylation and Expression in Proximal Colonic Mucosa

Cyclooxygenase-2 is a recognized risk factor in intestinal inflammation and CRC [31,32]. Levels of proximal colonic mucosal *Cox-2* mRNA were increased ~1.5-fold in mice fed the *n*-6HFD (Figure 3A). These results were consistent with other studies showing increased COX-2 expression associated with intestinal inflammation induced by an *n*-6HFD [33] and carcinogenesis of the colon [34,35]. Turning to the epigenetic regulation of the *Ptsg-2* promoter (Figure 3B), feeding of the *n*-6HFD to C57BL/6J mice reduced CpG methylation on average by ~35% (Figure 3B,C).

The gain of *Cox-2* mRNA levels and reduction in *Ptsg-2* CpG methylation were paralleled by an accumulation of COX-2 protein (Figure 4A). As a positive control for activation of COX-2 expression by the *n*-6HFD, we performed Western blotting of cell lysates from nontumor human FHC cells treated in culture with LA (Figure 4B), which induced COX-2 expression compared to control DMEM.

Taken together, these data suggest that exposure to an *n*-6HFD rich in LA elicits an inflammatory response characterized by CpG hypomethylation of *Ptsg-2* and increased expression of COX-2.

### 3.3. n-6HFD Lowers the Expression of Apc and Activates Downstream Targets for the β-Catenin/Wnt Pathway

The loss of *APC* along with the activation of the *KRAS* oncogene contribute to nuclear localization of β-catenin, a component of the Wnt signaling pathway, which promotes the expression of genes involved in proliferation [36,37,38]. An association between increased nuclear levels of β-catenin and COX-2 expression is seen in human and murine colon cancer cells with defective *Apc* [24,39]. We noted that the *n*-6HFD increased *Apc* CpG methylation (Figure 5A) while reducing the *Apc* transcript levels (Figure 5B) in colonic mucosa. Because the *Ccnd1* gene is a direct transcriptional target for activation by β-catenin transcription complexes [40,41], we measured the levels of *Ccnd1* transcripts and found increased expression associated with the *n*-6HFD (Figure 5C). The upregulation of c-JUN, whose gene is also a direct target for transcriptional activation by β-catenin [42] and contributes to the activation of COX-2 [43], provided a positive control for activation of the Wnt signaling pathway under conditions of diminished *Apc* expression (Figure 5D). Finally, we noted that the *n*-6HFD reduced the colonic levels of histone deacetylase-1 (HDAC-1) while increasing the expression of acetylated histone 3 (ACH3), suggesting effects on epigenetic regulation.

In summary, these results suggest that an *n*-6HFD rich in LA epigenetically alters gene expression in colonic mucosa, leading to an accumulation of proinflammatory and proliferative factors associated with a higher risk of CRC.

## 4. Discussion

In this study, we first address whether or not changes in *Fxr* promoter CpG methylation contribute to the regulation of *Fxr* expression in the colonic mucosa in response to an *n*-6HFD, which mimics dietary fat exposure known to increase the risk of intestinal inflammation in humans [3,4]. The total energy as fat (44%E) of the *n*-6HFD approaches the one used in previous mouse models that linked inflammation to the development of CRC [44,45]. We show that feeding of the *n*-6HFD to C57BL/6 mice reduces *Fxr* promoter CpG methylation while increasing the expression of *Fxr* and that of the FXR-target genes *Ibabp* and *Shp*. These results suggest that conditions that have been demonstrated to increase the intestinal levels of BA [9,10,30] direct the removal of repressive methylation marks at the *Fxr* gene to augment FXR expression and maintain BA homeostasis. We also show that the *n*-6HFD induces *Pparγ1* expression in the colonic mucosa. The upregulation of *Pparγ1* by the *n*-6HFD is consistent with the results of previous investigations documenting transactivation of the *Pparγ* gene by FXR through its physical interaction with an FXR response element [29], and genetic evidence that PPARγ expression is compromised in FXR-deficient (*Fxr*^−/−^) models [46]. The stimulatory action of FXR on the *PPARγ* gene is believed to reduce the steady-state levels of β-catenin in intestinal cells with normal *Apc* [47,48]. In addition, we report that the *n*-6HFD increases COX-2 expression associated with CpG hypomethylation at the *Ptsg-2* gene. In esophageal and gastrointestinal cells, BA increase the expression of COX-2 through various signals, including the p38-mitogen-activated protein kinase (p38MAPK) pathway [49] and the transcription factors nuclear factor kappa-light-chain-enhancer of activated B (NFkB) [50] and activator protein-1 (AP-1) [49,51]. Interestingly, SHP, a transcription factor whose expression is induced by FXR, interacts physically with and functions as a positive coactivator of NFkB [52], and is required for caudal-related homeobox-1 gene (*CDX1*)-mediated activation of COX-2 [53]. Therefore, the activation of FXR associated with chronic exposure to higher levels of BA due to the *n*-6HFD may have the adverse effect of triggering an inflammatory response supported by SHP, which in turn may override the anti-inflammatory activities of FXR on NFkB [54]. Moreover, the increased COX-2 expression is expected to support the enzymatic conversion of arachidonic acid (AA) to prostaglandin E2 (PGE2), which, through its EP2 and EP4 receptors, may activate the phosphoinositide 3-kinase (PI3K)/protein kinase B (Akt)-dependent upregulation of β-catenin. This may lead to further activation of COX-2 expression [55,56]. Thus, the long-term exposure to an *n*-6HFD rich in LA may trigger an intestinal feed-forward loop through which BA amplifies *COX-2* transcription mediated by the transcription factors β-catenin, SHP, and AP-1. In support of this hypothesis, we also show that the *n*-6HFD increases the expression of c-JUN, a key component of the AP-1 transcription factor [57]. Based on this cumulative evidence, we propose that accumulation of SHP and c-JUN coupled to CpG hypomethylation at the *Ptsg-2* gene may serve as a biomarker of increased risk of intestinal inflammation and tumorigenesis related to chronic exposure to an *n*-6HFD [58,59]. In accord with this suggestion, changes in DNA methylation profiles of genes involved in lipid metabolism and inflammation have been proposed as candidate biomarkers of CRC [14].

A second question addressed by this study pertains to whether or not feeding an *n*-6HFD epigenetically alters the expression of *Apc* and β-catenin signaling in the colonic mucosa. The rationale for this objective stems from earlier experimental evidence showing that colonic inflammation due to an HFD is paralleled by an increase in the gastro-intestinal levels of β-catenin [60,61,62] and the activation by the β-catenin/transcription factor (TCF)/lymphoid enhancer binding factor (LEF) transcription complex of genes (e.g., *Ccnd1*) involved in intestinal tumorigenesis [45,63,64]. We show that the CpG hypermethylation of the *Apc* gene in colonic mucosa of mice fed the *n*-6HFD correlates with reduced *Apc* expression, and an accumulation of *Ccnd1* and c-JUN, which are established transcription targets for the β-catenin/TC/LEF transcription complex [40,41]. The latter is constitutively active in *Apc*-deficient colon carcinoma [65,66,67]. One question raised by our data relates to the mechanisms contributing to the silencing of *Apc* in colonic mucosa of mice fed the *n*-6HFD. The enrichment in c-JUN and downregulation of *Apc* expression correlate in this study with a reduction in HDAC-1 and a gain in AcH3. These changes underscore the impact of dietary conditions that promote obesity, such as the weight gain observed in this study, on epigenetic regulation in the colonic mucosa [68]. Obesity induces DNA hypermethylation of *Apc* in the small intestine and changes the epigenetic landscape in the colonic epithelium [69], thus enhancing proliferation mediated by the Wnt/β-catenin pathway [70,71,72]. Studies in patients with gastric adenomas [73] and CRC cell lines [74] have confirmed the existence of a positive association between an accumulation of β-catenin and hypermethylation of the *Apc* gene. Recently, ~69% of CRCs were reported to harbor hypermethylated *Apc* [75], which correlated positively with tumor size and lymph node metastasis. To our knowledge, this is the first report showing a direct effect of an *n*-6HFD on epigenetic disregulation of *Apc* and *Pstg-2* via CpG methylation in the colonic mucosa. These epigenetic modifications may be useful in monitoring the susceptibility to CRC associated with the adoption of an *n*-6HFD and overweight conditions.

## 5. Conclusions

In summary, the current study provides evidence that an *n*-6HFD contributes epigenetically to the activation of *FXR* expression via CpG demethylation to support the expression of genes whose products (i.e., SHP, IBABP) participate in the regulation of BA homeostasis through the enterohepatic circulation (Figure 6). Our data also suggest that chronic exposure to an *n*-6HFD downregulates *APC* expression via CpG hypermethylation and this associates with increased expression of COX-2 via *PTSG-2* CpG hypomethylation and accumulation of C-JUN and *CCND1*, thus increasing the risk of inflammation and cancer of the colon.

Studies are ongoing in our laboratory to clarify the impact of diets that vary in their fatty acid profile (i.e., *n*-6 versus *n*-3) on CpG methylation of the *Fxr* and *Cox-2* genes and effects on the expression of FXR- and β-catenin target genes. Future investigators should elucidate how an *n*-6HFD modifies the interplay between microbiota and epigenetics of inflammation and colon cancer. For example, a diet mimicking the human Mediterranean diet, which is rich in olive oil, fruits, and vegetables, favors a microbiota composition associated with reduced carcinogenesis in intestinal cells with defective *Apc* [76]. Progress in these areas may help to address the question of whether the adoption of an *n*-6HFD associates with a higher risk of IBD and CRC and provide new strategies for epigenetic targeting through dietary interventions.

## Figures and Tables

**Figure 1 nutrients-11-00171-f001:**
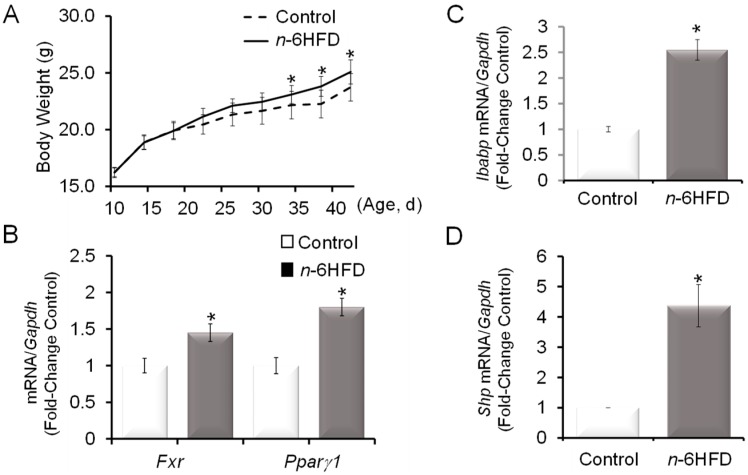
An *n*-6HFD (diet high in *n*-6 fatty acids) increases body weight and increases the expression of farnesoid-X-receptor (*Fxr*) in mouse colonic mucosa. (**A**) Weaned C57/BL6 male mice (*n* = 6) were assigned to a control diet containing 13% energy (13%E, 5% by weight safflower oil (SO), 76% linoleic acid (LA)) or an *n*-6HFD containing 44 energy (44%E, 22% by weight SO) for 6 weeks. Bars represent means ± standard error of the mean (SEM) of quantitation (fold change of control) of (**B**) *Fxr* and peroxisome proliferator-activated receptorγ1 (*Pparγ1*), (**C**) ileal bile acid-binding protein (*Ibabp*), and (**D**) small heterodimer protein (*Shp*) mRNA corrected for glyceraldehyde dehydrogenase phosphate (*Gapdh*) mRNA as an internal standard. Means ± SEM (standard error of the mean) with an asterisk (*) differ (*p* < 0.05).

**Figure 2 nutrients-11-00171-f002:**
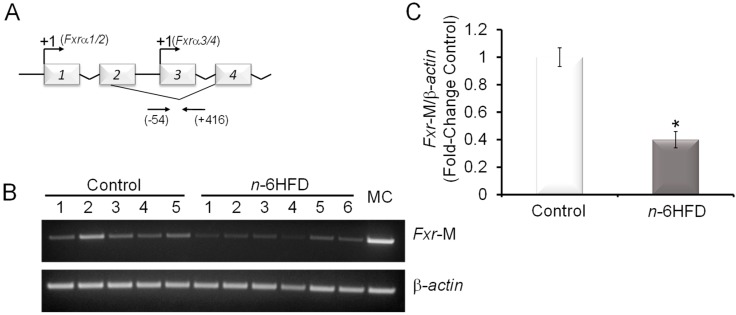
An *n*-6HFD induces *Fxr* CpG demethylation in mouse colonic mucosa. (**A**) Organization of the mouse *Fxr* gene. The top arrows indicate transcription start sites (+1) on exon-1 and exon-3. The bottom arrows indicate positions of oligonucleotides (−54/+416) around exon-3 used for CpG methylation studies [25]. (**B**) PCR bands amplified from bisulfonated genomic DNA obtained from proximal colonic mucosa with mouse *Fxr*- and *β-actin*-methylation (M)-specific primers. MC = methylation control. (**C**) Quantitation (fold-change/control) of *Fxr* promoter methylation status with control (*n* = 5) and *n*-6HFD (*n* = 6). Means ± SEM with an asterisk differ (*p* < 0.05).

**Figure 3 nutrients-11-00171-f003:**
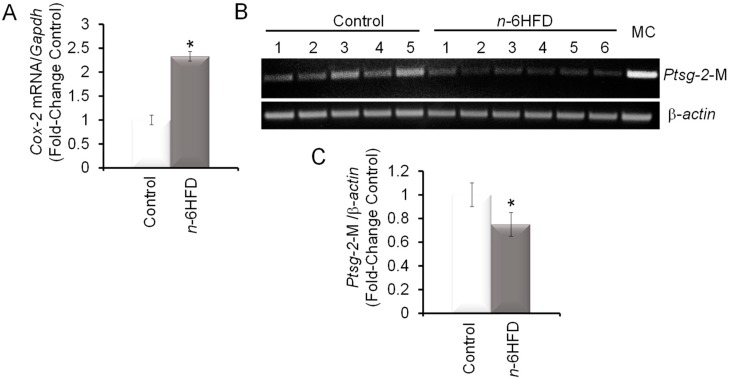
An *n*-6HFD induces *Ptsg-2* (prostaglandin-endoperoxide synthase-2) CpG demethylation in mouse colonic mucosa. (**A**) The bars represent means ± SEM of quantitation (fold change of control) of *Cox-2* mRNA corrected for *Gapdh* mRNA as an internal standard. Means ± SEM with an asterisk differ (*p* < 0.05). (**B**) PCR bands amplified from bisulfonated genomic DNA obtained from proximal colonic mucosa with mouse *Ptsg-2*- and *β-actin*-methylation (M)-specific primers. MC = methylation control. (**C**) Quantitation (fold-change/control) of *Ptsg-2* promoter methylation status with control (*n* = 5) and *n*-6HFD (*n* = 6).

**Figure 4 nutrients-11-00171-f004:**
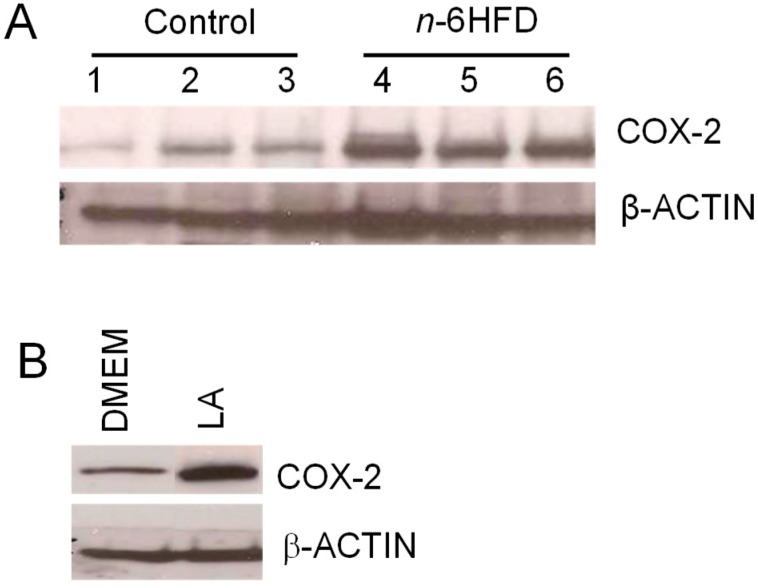
An *n*-6HFD induces COX-2 expression in mouse colonic mucosa. (**A**) The bands are representative immunocomplexes for COX-2 and internal control β-ACTIN in colonic mucosa of mice fed a control or an *n*-6HFD. (**B**) The bands are representative control immunocomplexes for COX-2 and internal control β-ACTIN from two separate experiments performed in triplicate in cell lysates of human fetal cells (FHC) cultured in control DMEM (Dulbecco’s Modified Eagle’s Medium) or DMEM plus LA (linoleic acid) (75 μM) for 72 h.

**Figure 5 nutrients-11-00171-f005:**
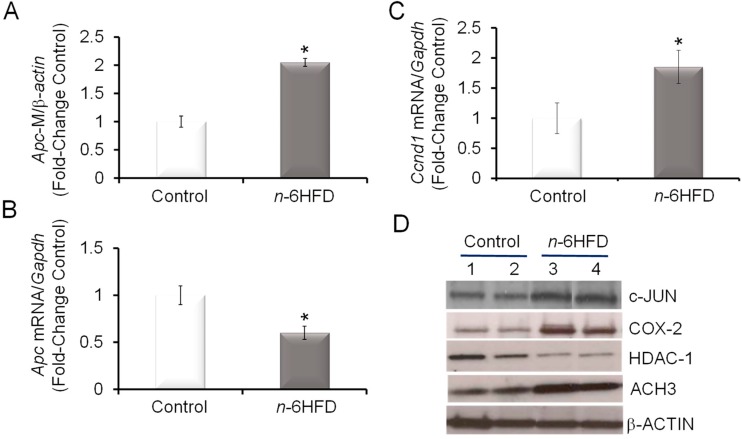
An *n*-6HFD induces *Apc* CpG hypermethylation in mouse colonic mucosa. (**A**) Quantitation (fold-change/control) of *Apc* promoter methylation status with control (*n* = 5) and *n*-6HFD (*n* = 6). (**B**) and (**C**) The bars represent respectively means ± SEM of quantitation (fold change of control) of *Apc* and *Ccnd1* mRNA corrected for *Gapdh* mRNA as an internal standard. Means ± SEM with an asterisk differ (*p* < 0.05). (**D**) The bands are representative immunocomplexes performed in duplicate for c-JUN, COX-2, histone deacetylase-1 (HDAC-1), acetylated histone 3 (ACH3), and internal control β-ACTIN in colonic mucosa of mice fed a control or *n*-6HFD.

**Figure 6 nutrients-11-00171-f006:**
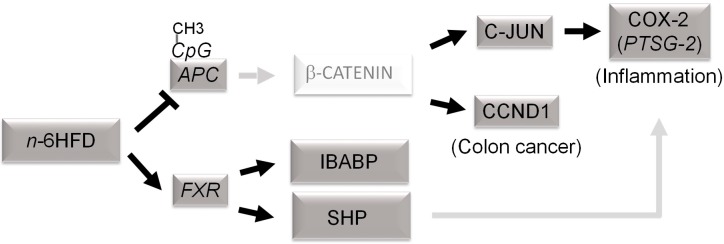
Proposed epigenetic model of colonic inflammation and carcinogenesis associated with long-term exposure to a diet high in *n*-6 fatty acids (*n*-6HFD). Chronic exposure to an *n*-6HFD impairs adenomatous polyposis Coli (*APC*) expression through CpG hypermethylation (CpG-CH_3_). This favors the activation of downstream targets of the β-catenin/Wnt pathway, supporting increased expression of genes involved in inflammation (c-JUN, *PTSG-2*) and proliferation such as cyclin D1 (CCND1). The factors in dark grey boxes refer to changes in expression from the current study. The involvement of β-catenin and upregulation of cyclooxygenase-2 (COX-2) by small heterodimer protein (SHP) are suggested based on published studies mentioned in the Discussion. FXR = farnesoid-X-receptor; IBABP = ileal bile acid-binding protein.

**Table 1 nutrients-11-00171-t001:** Diet composition ^a^.

Diet	Control		*n*-6HFD	
Formula	(g/Kg)		(g/Kg)	
Casein	200.0		240.0	
L-Cystine	3.0		3.6	
Corn Starch	397.5		199.4	
Maltodextrin	132.0		150.0	
Sucrose	120.0		80.0	
Safflower Oil	50.0		220.0	
Cellulose	50.0		50.0	
Mineral Mix, AIN-93G-MX(94046)	35.0		42.0	
Mineral Mix, AIN-93-VX(94047)	10.0		12.0	
Choline Bitartrate	2.5		3.0	
TBHQ, Antioxidant	0.01		0.045	
**Nutrient Composition**	(% Weight)	(% Kcal)	(% Weight)	(% Kcal)
Protein	17.7	19.3	21.2	18.7
Carbohydrate	62.1	67.9	42.3	37.3
Fat	5.2	12.8	22.2	44.0
**Energy (Kcal/g)**	3.7		4.5	

^a^ Values are calculated from ingredient analysis or manufacturer data (Harlan Laboratories). *n*-6HFD = diet high in *n*-6 fatty acids; TBHQ = Tertiary butylhydroquinone.

**Table 2 nutrients-11-00171-t002:** The primers for RT-PCR and CpG methylation mouse studies.

Target	Primer Sequence ^a^
**mRNA:**	
*Fxr*	F: TTAGTCTTCACCACAGCCACC
	R: ACCTGTATACATACATTCAGCCAAC
*Apc*	F: CTGAGCCTGGATGAGCCATT
	R: GTGAGTCCAAGGCGAACGTC
*Pparγ1*	F: GTGAGACCAACAGCCTGACG
	R: ACAGACTCGGCACTCAATGG
Cox-2	F: GAAGTCTTTGGTCTGGTGCCT
	R: GCTCCTGCTTGAGTATGTCG
*Gapdh*	F: CACTTGAAGGGTGGAGCCAA
	R: AGTGATGGCATGGACTGTGG
*Ibabp*	F: CAGGAGACGTGATTGAAAGGG
	R: GCCCCCAGAGTAAGACTGGG
*Shp*	F: GTACCTGAAGGGCACGATCC
	R: AGCCTCCTGTTGCAGGTGT
*Ccnd1*	F: CTAAACAAGCACCCCCTCCA
	R: GGTAACAGGGCTGTAGGCAC
**Methylation-specific:**	
*Fxr*	F: CGTTTAGCGATGGGGTTAATTAG
	R: CGTCTTCTTTACTTATCTAAACCTCCTT
*Apc*	F: GAGTGTGGTTGTCGGAAATTC
	R: CAAAAAAACGTACATAAAAAACGCT
*Ptsg-2*	F: TTTTAGTTAGGATTTTAGATTTCGG
	R: ATAATACCAAAAAAACTACACCGC
*β-Actin*	F: AATAGTTATTTTAAGTATTTATGAAATAAG
	R: TAACTACCTCAACACCTCAAC

^a^ F = forward; R = reverse. *Apc* = adenomatous polyposis Coli; *Cox-2* = cyclooxygenase-2; *Ccnd1* = cyclin D1; *Fxr* = farsenoid-X-receptor; *Gadph* = glyceraldehyde dehydrogenase phosphate; *Ibabp* = ileal bile acid-binding protein; *Pparγ1* = peroxisome proliferator-activated receptorγ1; *Ptsg-2* = prostaglandin-endoperoxide synthase-2; *Shp* = small heterodimer protein.

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
