# Peer review of "n-6 Linoleic Acid Induces Epigenetics Alterations Associated with Colonic Inflammation and Cancer"

_nutrients, 2019, doi:10.3390/nu11010171_

Round 1

Reviewer 1 Report

This study focuses on the molecular mechanism by which the n-6 PUFA linoleic acid in the diet promotes colitis and colon cancer. The authors fed mice with a diet containing high levels of linoelic acid chronically and then assessed the expression of various genes related to enterohepatic circulation of bile salts and also cancer-associated WNT signaling pathway. The central theme of the study appears to be the idea that exposure to linoleic acid changes the epigenetic profile (i.e., DNA methylation) of colonic epithelial cells. The data do show that the promoter methylation of three genes, namely Fxr, Apc, and Cox2, are altered in linoleic acid-exposed colon. Interestingly, the methylation is decreased in one case, but increased in two cases. These changes correlate well with the mRNA levels with the idea that hypomethylation enhances gene expression whereas hypermethylation silences gene expression. What is missing however is an explanation as to how linoleic acid elicits these differential effects on these genes. The authors never raised this question and never addressed it even in the discussion. But this is an important issue that needs explaining.

The authors summarize the data in the form a model in Fig. 7. But it is somewhat confusing, considering the data presented in the manuscript. First of all, why do the authors talk about acute exposure versus chronic exposure when all the animal studies apparently represent chronic exposure. Obviously, the studies presented in the manuscript are not related in any way to acute exposure. Secondly, the authors highlight a couple of references in the legend to Fig. 7; it is not clear whether the ideas indicated in the figure have already been published in these references or the references are given for some other purpose, which is not immediately clear to the reader.

The authors show that linoleic acid enhances Fxr, correlating with increased expression of its target genes. This would mean that enterohepatic circulation of bile acids is efficient in linoleic acid-exposed mice. This would decrease the levels of DCA in colon. But Fig. 7 notes that DCA levels go up in colon in response to linoelic acid treatment.

The data in Fig. 6 show that treatment of colon cells with DCA increases APC; this means that linoleic acid exposure, which decreases DCA levels in colon, must decrease APC expression. This interpretation is consistent with the idea that linoleic acid promotes colon cancer. But what does this do to the concept that increased levels of bile acids in the large intestine promotes cancer? If this is true, linoleic acid which decreases the levels of DCA in colon, must protect against colon cancer, a conclusion quite the opposite of what linoleic acid is proposed to do.

Literature evidence points out that FXR might be a tumor suppressor whereas COX2 is a tumor promoter. Linoleic acid increases FXR expression and signaling; how does this relate to the tumor-promoting role of this fatty acid? COX2 is a tumor promoter, and linoelic acid increases its expression. This is okay in relation to the tumor-promoting role of the fatty acid.    

Author Response

1.     What is missing however is an explanation as to how linoleic acid elicits these differential effects on these genes. The authors never raised this question and never addressed it even in the discussion. But this is an important issue that needs explaining.

We have modified the discussion to address the question as to how a diet rich in linoleic acid exerts differential effects of various genes. We have included new references to support the discussion.

2.     The authors summarize the data in the form a model in Fig. 7. But it is somewhat confusing, considering the data presented in the manuscript. First of all, why do the authors talk about acute exposure versus chronic exposure when all the animal studies apparently represent chronic exposure. Obviously, the studies presented in the manuscript are not related in any way to acute exposure.

We have removed the reference to the acute exposure and renumbered Figure 7 to Figure 6 (previous Figure 6 was removed based on the suggestion of reviewer 2).

3.     Secondly, the authors highlight a couple of references in the legend to Fig. 7; it is not clear whether the ideas indicated in the figure have already been published in these references or the references are given for some other purpose, which is not immediately clear to the reader.

We have modified the Figure (now #6) and the legend now refers to data originated in this study and published previously mentioned in the Discussion.

4.     The authors show that linoleic acid enhances Fxr, correlating with increased expression of its target genes. This would mean that enterohepatic circulation of bile acids is efficient in linoleic acid-exposed mice. This would decrease the levels of DCA in colon. But Fig. 7 notes that DCA levels go up in colon in response to linoleic acid treatment.

We included in the Discussion a suggestion that chronic exposure to LA may override the ability of FXR to direct efficient BA homeostasis in spite of the increased expression of FXR. Figure 7 (now Figure 6) has been revised.

5.     The data in Fig. 6 show that treatment of colon cells with DCA increases APC; this means that linoleic acid exposure, which decreases DCA levels in colon, must decrease APC expression. This interpretation is consistent with the idea that linoleic acid promotes colon cancer. But what does this do to the concept that increased levels of bile acids in the large intestine promotes cancer? If this is true, linoleic acid which decreases the levels of DCA in colon, must protect against colon cancer, a conclusion quite the opposite of what linoleic acid is proposed to do.

Figure 6 has been removed based on suggestion of reviewer#2.

6.     Literature evidence points out that FXR might be a tumor suppressor whereas COX2 is a tumor promoter. Linoleic acid increases FXR expression and signaling; how does this relate to the tumor-promoting role of this fatty acid? COX2 is a tumor promoter, and linoleic acid increases its expression. This is okay in relation to the tumor-promoting role of the fatty acid.    

We mention in the revised Discussion that the upregulation of FXR as a result of the HFD is an adaptive response to maintain BA homeostasis, however, we suggest that this does not preclude upregulation of COX-2 and inflammation due to chronic exposure to the HFD.

Reviewer 2 Report

Romagnolo et al. present for publication on Nutrients a manuscript describing the epigenetics alterations occurring, in vivo, upon chronic consumption of a diet enriched in n-6 linoleic acid (LA). LA is a very abundant n-6 found in plant oil. Diets high in n-6 fatty acids (n-6HFD) are recognized risk factors for inflammatory bowel diseases and ulcerative colitis. LA is known to 1) induce cell proliferation in the colon, 2) increase fecal levels of bile acids and 3) increase colon cancer incidence in several biological systems.

In an in vivo murine model (C57/BL6 mice) the authors prove by qPCR that n-6HFD increases the transcription of the genes coding for FXR and COX2. For both genes, this increase seems to be ascribed to hypomethylation of their promoters, one of the consequences of a n-6FHD.

Differently a n-6HFD lowers the expression of APC gene, a tumor suppressor involved in b-catenin/Wnt pathway and colon cancer progression. The decreased expression of APC results from hypermethylation of the APC promoter. Low levels of APC result in the expression of Wnt pathway gene targets c jun and cyclin D.

The manuscript is technically sound. The experiments performed to analyze the methylation state of the three gene-promoters are well performed and the results are clear. However, the final attempt to connect all the results to each other is confusing and this, overall, results in a discussion (and especially in a final cartoon model described in figure 7) that is simplistic and chaotic. I thus believe that the manuscript must be slightly improved and requires a revision.

The huge problem here is Figure 7. Which experiments are summarized by the dashed lines? In the legend is written “acute exposure”. However, acute exposure is poorly or never discussed in the text. This does not help the readers. Do dashed lines summarize the experiments described in Figure   6 (upregulation of APC and FXR in cells treated with the bile acid DCA)? Or we should look at references 48 and 25 to understand these dashed lines? (was reference 25 published by this lab as stated at line311?)

In the first case( i.e. dashed lines summarize experiments in HCT-116 cells) which experiments describes the change in methylation of APC promoter in in vitro cultured HCT-116 cells? Where is shown that LA reduces COX2 expression in HCT-116? (Actually, Figure 4B shows that LA induces COX2 expression in FHC). Where is shown that this event results from modulation of ACH3? Moreover, I do not think that in vitro experiments can summarize well in vivo acute exposure to LA.

In the second case ( i.e. dashed lines summarize data in the literature describing the effect exerted by  LA) this reviewer thinks that references, here, are not appropriate. This is not a review and the cartoon should summarize the content of the article and not those present in other manuscripts.

In both cases, the dashed lines here in the cartoon are worsening the all manuscript.

On the contrary, the solid lines could describe the actual in vivo experiments present in this manuscript and are much more clear. The only remark would be that the increase in stability of beta-catenin should be the result of APC inhibition and not a direct result of n-6HFD (see discussion lines 304-306). A solid line should indicate this loss of inhibition upon APC inactivation by n-6HFD. I also think that it is improper to include beta-catenin, c-jun in the same box of DCA. Is not DCA the result of the conversion of a primary bile acid (produced by hepatocytes) by colon microbiota, upon feeding a high-fat diet?

My request is

i) to keep in vivo data and remove experiments using siAPC HCT116 cells.  

ii) to remove dashed lines in the cartoon of figure 7 and focus on the in vivo data that are interesting and clear.

iii) The discussion should be rephrased accordingly.

Minor point:

 Legend to Figure 6 : the number 72 misses the h of hours.

Author Response

The final attempt to connect all the results to each other is confusing and this, overall, results in a discussion (and especially in a final cartoon model described in figure 7) that is simplistic and chaotic. I thus believe that the manuscript must be slightly improved and requires a revision.

The Discussion has been revised to address these concerns.

The huge problem here is Figure 7. Which experiments are summarized by the dashed lines? In the legend is written “acute exposure”. However, acute exposure is poorly or never discussed in the text. This does not help the readers. Do dashed lines summarize the experiments described in Figure   6 (upregulation of APC and FXR in cells treated with the bile acid DCA)? Or we should look at references 48 and 25 to understand these dashed lines? (was reference 25 published by this lab as stated at line311?)

Figure 7, (now Figure 6) has been modified to remove reference to acute exposure and provide a better picture based on new information included in the revised Discussion.

In the first case( i.e. dashed lines summarize experiments in HCT-116 cells) which experiments describes the change in methylation of APC promoter in in vitro cultured HCT-116 cells? Where is shown that LA reduces COX2 expression in HCT-116? (Actually, Figure 4B shows that LA induces COX2 expression in FHC). Where is shown that this event results from modulation of ACH3? Moreover, I do not think that in vitro experiments can summarize well in vivo acute exposure to LA.

In vitro experiments with HCT-116 cells have been removed.

In the second case ( i.e. dashed lines summarize data in the literature describing the effect exerted by  LA) this reviewer thinks that references, here, are not appropriate. This is not a review and the cartoon should summarize the content of the article and not those present in other manuscripts.

In both cases, the dashed lines here in the cartoon are worsening the all manuscript.

Thanks for the suggestions. We have revised the figure (now Fig. 6)

On the contrary, the solid lines could describe the actual in vivo experiments present in this manuscript and are much more clear. The only remark would be that the increase in stability of beta-catenin should be the result of APC inhibition and not a direct result of n-6HFD (see discussion lines 304-306). A solid line should indicate this loss of inhibition upon APC inactivation by n-6HFD. I also think that it is improper to include beta-catenin, c-jun in the same box of DCA. Is not DCA the result of the conversion of a primary bile acid (produced by hepatocytes) by colon microbiota, upon feeding a high-fat diet?

My request is

i)                to keep in vivo data and remove experiments using siAPC HCT116 cells.  

Done.

ii)               to remove dashed lines in the cartoon of figure 7 and focus on the in vivo data that are interesting and clear.

Done

iii)              The discussion should be rephrased accordingly.

Done

Minor point:

 Legend to Figure 6 : the number 72 misses the h of hours.

Figure 6 with siAPC data from HCt-116 has been deleted based on the above suggestion.

Round 2

Reviewer 1 Report

None. The authors have tried their best to address my concerns raised during the first review.

Reviewer 2 Report

I am satisfied with the revised version of  manuscript.